# Density-Dependent Differentiation of Tonsil-Derived Mesenchymal Stem Cells into Parathyroid-Hormone-Releasing Cells

**DOI:** 10.3390/ijms23020715

**Published:** 2022-01-10

**Authors:** Ji Yeon Kim, Saeyoung Park, Se-Young Oh, Yu Hwa Nam, Young Min Choi, Yeonzi Choi, Ha Yeong Kim, Soo Yeon Jung, Han Su Kim, Inho Jo, Sung-Chul Jung

**Affiliations:** 1Department of Biochemistry, College of Medicine, Ewha Womans University, Seoul 07804, Korea; staramom@gmail.com (J.Y.K.); saeyoung@ewha.ac.kr (S.P.); queennnam@ewha.ac.kr (Y.H.N.); cyz72@naver.com (Y.C.); 2Departments of Molecular Medicine, College of Medicine, Ewha Womans University, Seoul 07804, Korea; ohs@ewha.ac.kr (S.-Y.O.); cymin717@gmail.com (Y.M.C.); inhojo@ewha.ac.kr (I.J.); 3Graduate Program in System Health Science and Engineering, Ewha Womans University, Seoul 07804, Korea; 4Departments of Otorhinolaryngology-Head and Neck Surgery, College of Medicine, Ewha Womans University, Seoul 07804, Korea; ha0@ewha.ac.kr (H.Y.K.); mdjungsy@ewha.ac.kr (S.Y.J.); sevent@ewha.ac.kr (H.S.K.)

**Keywords:** tonsil-derived mesenchymal stem cells, parathyroid-hormone-releasing cells, differentiation, contact inhibition, donor-dependent variation

## Abstract

Mesenchymal stem cells (MSCs) can differentiate into endoderm lineages, especially parathyroid-hormone (PTH)-releasing cells. We have previously reported that tonsil-derived MSC (T-MSC) can differentiate into PTH-releasing cells (T-MSC-PTHCs), which restored the parathyroid functions in parathyroidectomy (PTX) rats. In this study, we demonstrate quality optimization by standardizing the differentiation rate for a better clinical application of T-MSC-PTHCs to overcome donor-dependent variation of T-MSCs. Quantitation results of PTH mRNA copy number in the differentiated cells and the PTH concentration in the conditioned medium confirmed that the differentiation efficiency largely varied depending on the cells from each donor. In addition, the differentiation rate of the cells from all the donors greatly improved when differentiation was started at a high cell density (100% confluence). The large-scale expression profiling of T-MSC-PTHCs by RNA sequencing indicated that those genes involved in exiting the differentiation and the cell cycle were the major pathways for the differentiation of T-MSC-PTHCs. Furthermore, the implantation of the T-MSC-PTHCs, which were differentiated at a high cell density embedded in hyaluronic acid, resulted in a higher serum PTH in the PTX model. This standardized efficiency of differentiation into PTHC was achieved by initiating differentiation at a high cell density. Our findings provide a potential solution to overcome the limitations due to donor-dependent variation by establishing a standardized differentiation protocol for the clinical application of T-MSC therapy in treating hypoparathyroidism.

## 1. Introduction

The parathyroid gland is an endocrine organ that dynamically secretes parathyroid hormone (PTH) as a polypeptide consisting of 84 amino acids, in response to changes in extracellular calcium concentrations, regulating ion homeostasis [1]. The most common etiology of hypoparathyroidism is post-thyroidectomy hypoparathyroidism since the parathyroid gland is located adjacent to the thyroid gland [2]. The incidence of hypoparathyroidism has increased along with the increasing incidence of thyroid surgery. Hypoparathyroidism causes low serum calcium and high serum phosphorus levels, as well as inadequate/deficient PTH secretion [2,3]. The most common treatment for this condition is hormone replacement therapy with calcium and vitamin D, but this may not be sufficient to compensate for the loss of endocrine function. Although glandular autograft is known to be the most effective technique, in place of alternative therapies, autologous transplantation is not always possible. Allografts are a good way to treat patients who are unable to receive autograft treatment, but this has limited success due to the side effects associated with tissue rejection [4]; therefore, MSC therapy as a treatment for hypoparathyroidism is expected to be used clinically in the future.

Mesenchymal stem cells (MSCs) are pluripotent cells derived from various tissues and have the ability to differentiate into various cell types of the three germ layers and have therapeutic properties [5]. Tonsil tissue is a newly identified source of MSCs that have potential therapeutic applications. Tonsil-derived MSCs (T-MSC) have the capacity to differentiate into cells of various lineages, including mesodermal (fat, cartilage, and bone cells), endodermal (hepatocytes and parathyroid-like cells), and ectodermal (neuron) lineages [6,7]. In addition to their unique differentiation and proliferation potentials, T-MSCs exhibit immunosuppressive properties similar to those of bone-marrow-derived MSCs and adipose-tissue-derived MSCs [8,9]. Isolation of T-MSCs from tonsil tissues obtained from tonsillectomies makes them easily available, and this provides a valuable means of recycling human tissue [9,10]. However, the proliferation rate and differentiation ability of T-MSCs showed donor-to donor variation. This donor-dependent variation of T-MSCs gives unexpected limitations to the preclinical and clinical development of T-MSCs. To accelerate the clinical application of T-MSCs that were demonstrated to differentiate into multiple lineages, it is necessary to optimize the quality of therapeutics through standardization of cell differentiation rates.

It has been reported that increased culture density is associated with decreased proliferation, prolonged G1 stage, and enhanced propensity for differentiation of self-renewing human pluripotent stem cells [11]. In the case of embryonic stem cells, they exhibit a high density of cells for embryonic body (EB) formation, an essential step for differentiation [12]. Becker et al. (2010) reported that the length of the G1 phase of human embryonic stem cell (hESC) culture is inversely proportional to the cell density upon plating, which has been demonstrated to possibly be due to the roles of autologous/paracrine factors and cell–cell contacts [13].

To find a way to standardize differentiation into T-MSC-PTHCs, differentiation was initiated at cell densities of 50% or 100% (low cell density or high cell density). Since different proliferation rates vary by donor, T-MSCs derived from three donors were studied. The characteristics of PTHCs were confirmed by measuring the expression of PTH gene and calcium-sensing receptor (CaSR) protein cells, and the concentration of PTH in a conditioned medium (CM). In addition, large-scale expression profiling of T-MSC-PTHCs by RNA sequencing was performed to investigate the differentiation mechanism related to the cell density. T-MSC-PTHCs with the highest differentiation rate were transplanted with hyaluronic acid (HA) scaffolds into parathyroidectomy (PTX) rats to confirm the treatment effects. Consequently, we investigated the possibility of clinical applications of human T-MSCs-PTHCs for parathyroid regeneration.

## 2. Results 

### 2.1. Differentiation into T-MSC-Derived PTH-Releasing Cells (PTHCs) according to Cell Density

To investigate the effect of cell density on differentiation into T-MSC-PTHCs, T-MSCs derived from three donors were, respectively, differentiated at two different cell densities: low (50% confluence) or high (100% confluence). The proliferation capacity of T-MSCs from three donors showed donor-to-donor variation. Therefore, the cell numbers to reach 50 and/or 100% confluency were different for each donor. The scheme in Figure 1A illustrates the protocol for differentiation into T-MSC-PTHCs (D7). The approximate number of cells required to reach 50 or 100% confluency in one day for each donor is provided in the scheme shown in Figure 1A. As a result of measuring the PTH mRNA copy number of T-MSC-PTHCs using droplet digital PCR (ddPCR), the increase was greater when the differentiation was initiated in high-density cells (#1: 526 ± 742.1 copies; #2: 1310 ± 646.3 copies; #3: 1155 ± 1061 copies) than in low-density cells (#1: 3465 ± 776.9 copies; #2: 33.7 ± 397.2 copies; #3: 3613 ± 236.9 copies) in donors (#1–3). In addition, only donor #2 in the low cell-density group (Figure 1B), and donors #1–3 in the high-density cell group (Figure 1C) significantly increased the PTH mRNA copy number. The morphologies of the cells at D0 and D7 used in this study were observed by a light microscope (Figure 1D). The images of the T-MSCs from donor #1 at high cell density showed the most densely packed cells in the cell culture dish.

The high cell-density group from all donors had a higher differentiation efficiency than the low cell density in the ddPCR analysis, and the secreted PTH concentration and protein level of calcium-sensing receptor (CaSR) were measured using ELISA and Western blots, respectively. In the case of the T-MSCs at low cell density, PTH secretion was not significantly different between pre- and post-differentiation (data not shown). The PTH concentration increased only in all donors of high cell density after differentiation, but in particular, donors #1 and #2 showed a significant increase (*p* < 0.05 and *p* < 0.01, respectively) (Figure 2A). PTH concentrations of T-MSC in the CM of donors #1, #2, and #3 were 6.05 ± 0.40 pg/mL, 7.61 ± 2.02 pg/mL, and 10.43 ± 4.51 pg/mL, and T-MSC-PTHC was 12.21 ± 4.05 pg/mL, 20.31 ± 1.62 pg, and 15.27± 7.69 pg/mL, respectively. In a previous study, we confirmed an increase in PTH secreted from T-MSC-PTHCs differentiated into T-MSCs from one donor (Park YS, 2015). In addition, the iPTH secretion level of T-MSC-PTH observed in this study showed typical characteristics of PTHC when considering the physiological iPTH concentration in humans (15~65 pg/mL) (Youngwirth L, 2010). Additionally, we investigated the expression of CaSR, a marker of parathyroid development, and of mature parathyroid glands [14] using Western blotting. The expression of CaSR protein did not significantly increase at low cell density (Figure 2B) and increased by 2~5-fold in high cell density after differentiation, but only donors #1 and #3 showed significant changes (*p* < 0.001) (Figure 2C). These results demonstrate that contact inhibition by a high cell density is an important factor in the differentiation of PTHCs. The high cell-density group of donor#1 T-MSCs had the highest differentiation efficiency of all, so it was used alone in subsequent studies.

### 2.2. CaSR Protein Expression of T-MSC-PTHCs in Response to Extracellular Calcium Levels

As a result of immunocytochemistry, it was also found that the expression of PTH protein increased after differentiation into T-MSC-PTHC (Figure 3A,C), and the group with a high cell density (Figure 3B) had a higher expression level, compared with the group with a low cell density (Figure 3A). Since CaSR is related to extracellular calcium concentration-dependent PTH secretion, we investigated whether T-MSC-PTHC responds to CaSR expression in response to extracellular calcium level [15,16]. As a result, T-MSC-PTHCs exposed to low calcium concentrations (0.09 mM, Low Ca^2+^) for 48 h increased CaSR expression, compared with T-MSC and the high Ca^2+^ exposure group, in both cell densities (Figure 3B,D). The rate of increase in CaSR for T-MSC (1.0 ± 1.0) in the low Ca^2+^ exposure group was higher in the high cell-density group (7.03 ± 2.92; Figure 3D) than in the low cell-density group (5.10 ± 1.18; Figure 3B).

### 2.3. RNAseq Analysis by Gene Set Enrichment Analysis (GSEA)

The transcriptomes were analyzed using RNA sequencing (RNAseq), followed by the large-scale expression profiling of T-MSC-PTHCs on T-MSCs. We analyzed the signaling pathway in the T-MSC and T-MSC-PTHCs by using gene set enrichment analysis (GSEA). The top 20 most significantly upregulated and downregulated expressed genes of the T-MSC-PTHCs are shown in Appendix A, respectively. Then, 12 and 20 known genes related to cell cycle and differentiation were selected, respectively, which were significantly changed in T-MSC-PTHCs, compared with the undifferentiated T-MSCs (Table 1 and Table 2). Clusters of cell cycle and differentiation were found only within a minority of the T-MSC-PTHCs (Figure 4A). Changes in the expression of genes were involved in the active pathways of the T-MSC-PTHCs. For example, the genes involved in the negative regulation of chromatin organization, cell aggregation, and endodermal differentiation were significantly upregulated in the T-MSC-PTHC group (Figure 4B). Based on the Transcriptome Analysis Console (TAC) results, terms related to “signaling pathway regulating pluripotency of stem cells” for functional differentiated cells and “TGF-beta signaling” for G1 arrest of cell cycles in the KEGG pathways, Wnt, SMADs, BMP, and Smad1/5/8 were increased in T-MSC-PTHCs (Figure 4C,D).

### 2.4. Measurement of Serum PTH by T-MSC Transplantation in the PTX Mouse Model

To evaluate the therapeutic efficacy of T-MSC-PTHCs, we transplanted the cells mixed with hyaluronic acid (HA) at the same time as the preparation of PTX rat and measured serum intact PTH (iPTH) level in rats for 16 weeks. Compared with the non-PTX group, iPTH secretion was not improved in the T-MSC-HA and HA alone groups. However, the T-MSC-PTHC-HA group showed a tendency to increase in serum iPTH compared to the T-MSC-HA and the HA groups only throughout the experiment period. The range of serum iPTH levels in the T-MSC-PTHC-HA group was close to the average serum PTH level in humans (8~51 pg/mL) [17], although not as much as the non-PTX group (Figure 5). 

## 3. Discussion

In previous parathyroid replacement studies, we successfully demonstrated T-MSC differentiation into PTHCs using a PTX rat model. In addition, the differentiated T-MSC-PTHCs embedded in a Matrigel restored in vivo parathyroid function [18]. In another study, we demonstrated that T-MSC differentiated into parathyroid tissue spheroids and confirmed that the T-MSC-PTHC spheroids were highly viable (>80%); they expressed high levels of iPTH, the parathyroid secretory protein 1, and cell adhesion molecule, N-cadherin. Furthermore, the implantation of a spheroid form of the differentiated T-MSC in PTX rats resulted in higher survival rates (50%) over a 3-month period, with high physiological levels of both serum iPTH and ionized calcium, compared with PTX rats treated with either vehicle or undifferentiated T-MSC spheroids [19]. Nevertheless, clinical application for patients with hypoparathyroidism would be difficult because the cells to be transplanted did not have a standardized differentiation capacity due to donor variation. Donor-dependent variation in proliferation and differentiation of mesenchymal stem cells have been also observed in other mesenchymal stem cells from different tissue sources, such as bone marrow, adipose tissue, and umbilical cord blood [20,21,22]. Thus, we hypothesized that standardization of differentiation rates by high cell density could be the solution to overcoming this limitation.

Based on previous studies [18], we planned the differentiation period to be 7 days, and to demonstrate that high cell density is a key factor in increasing differentiation rate, we induced T-MSCs from three donors to differentiate into parathyroid-like cells at two cell densities (low or high). The results showed higher PTH mRNA expression levels in the high cell-density group, compared with the low cell-density group, in all donors. The cells seeded at high cell density expressed higher levels of PTH protein, compared with those seeded at low cell density. These findings are similar to previous studies, in which contact inhibition is dependent on cell density and a higher cell density increases intercellular communication, inducing cellular functionality [23,24]. A sensitive balance between a variety of inhibitors and activators of progression in the cell cycle regulated the switch between cell cycle arrest and cell proliferation [25]. Cell cycle arrest may result in one of several possible outcomes—namely, quiescence, senescence, apoptosis, motility, or differentiation. Based on these reports, it can be speculated that a protocol starting at 100% confluence (high cell density) could standardize the differentiation rate of the cells sourced from different donors.

CaSR in the parathyroid gland is important because CaSR-regulated PTH secretion plays a major role in regulating ion homeostasis in blood, altering renal function [26]. To confirm the characterization of T-MSC-PTHCs, we compared the presence of CaSR protein between undifferentiated and differentiated T-MSCs. We found that the expression of CaSR in T-MSC-PTHC was significantly increased, compared with T-MSC. It was found in this study that CaSR expression of T-MSC-PTHC was higher at low extracellular calcium concentrations than those at high calcium concentrations. These results suggest that T-MSC-PTHCs are regulated by extracellular calcium levels, suggesting their functionality as parathyroid-like cells as demonstrated from the findings of the previous studies [18,19].

Cell-cell interactions are important in regulating endocrine cell secretion [27]. Sun et al. (1994) suggested that parathyroid cells in close proximity are stimulated to secrete more hormones than those at lesser densities. In addition, parathyroid cells have a tendency to secrete PTH at a higher efficiency when plated at a high density [28]. Guo et al. (2009) provided evidence that differentiation licensing of the 3T3-L1 cells during the contact-inhibition stage involved epigenetic modification [29]. High-cell density/confluency leads to contact inhibition of proliferation, as seen in the majority of epithelial cells, and is associated with the initiation of differentiation [30]. In this regard, we analyzed transcriptomes using RNAseq, then performed a large-scale expression profiling of T-MSCs and performed a comparison between undifferentiated T-MSCs and T-MSC-PTHCs. RNAseq analysis showed that cell cycle and differentiation signal pathway were found only within a minority in T-MSC-PTHCs. In particular, GSEA results showed that negative regulation of chromatin organization, cell aggregation, and endodermal differentiation signaling pathway enriched with genes were upregulated in T-MSC-PTHCs. KEGG pathways analysis indicated that Wnt, SMADs, BMP, and Smad1/5/8 relating to pluripotency of stem cell pluripotency-regulating pathway and G1 arrest of the cell cycle in the TGF-beta signaling pathway, as well as endodermal differentiation, were increased in the T-MSC-PTHCs. 

Salwowska et al. (2016) reported that therapies with HA provided long-lasting, pain-relieving, moisturizing, lubricating, and dermal filling effects. In ophthalmology, the role of HA in lubricating the corneal endothelium is well established [31], and it has also been shown to improve tissue hydration and cellular resistance to mechanical damage in aesthetic dermal tissue [32]; however, the use of HA scaffolds with hormone-releasing cells have not been reported yet. In this study, the implantation of T-MSC-PTHC embedded in HA resulted in the slow release of PTH in PTX rats for at least 16 weeks, demonstrating the chemical properties of the HA could benefit hormone-releasing cells to be more successfully used in clinical trials.

PTH gene expression significantly increased at high cell densities than at low cell densities in all donors (#1~#3). iPTH concentration in CM was significantly increased in donors #1 and #2, and intracellular CaSR protein expression was significantly increased in T-MSC-PTHC from donors #1 and #3. The cells from donor #1 were selected for further experimental trials due to their ability to increase CaSR in low calcium, and this was even further increased when differentiated at high cell density. The mechanism of differentiation was also observed using large-scale expression profiling of the T-MSC-PTHCs by RNA sequencing. Furthermore, the T-MSC-PTHCs of donor #1 derived from a high cell density resulted in a better therapeutic effect in the PTX model. In this study, it was confirmed that an important factor for differentiation into parathyroid-like cells was the cell density, where a high cell density close to 100% confluence promotes cell differentiation by inducing cell-cell contact inhibition. Our findings provide a solution to control donor-dependent cellular variation to achieve constant levels of cell differentiation for the clinical application of T-MSCs therapy for treating hypothyroidism.

## 4. Materials and Methods

### 4.1. T-MSC Isolation and Culture

T-MSC isolation and culture were conducted as previously described [6,7,18]. Informed written consent was obtained from all patients participating in the study. The study protocol was approved by the Ewha Womans University Medical Center (EWUMC) institutional review board (IRB number: ECT-2011-09-003). To establish T-MSCs from tonsillar tissues, isolated tonsillar tissue was minced and digested in Dulbecco’s modified Eagle medium (DMEM; Invitrogen, Carlsbad, CA, USA) containing 210 U/mL collagenase type I (Invitrogen, Carlsbad, CA, USA) and 10 μg/mL DNase (Sigma-Aldrich, St. Louis, MO, USA) for 30 min at 37 °C. The digested tissue was filtered through a wire mesh and washed with RPMI-1640 medium, after which adherent mononuclear cells were obtained by Ficoll-Paque (GE Healthcare, Little Chalfont, UK) density gradient centrifugation. The cells were cultured for 48 h at 37 °C in high-glucose (4500 mg/L) DMEM containing 10% fetal bovine serum (FBS; Invitrogen) and 1% penicillin-streptomycin (Sigma-Aldrich, St. Louis, MO, USA) in a humidified chamber under 5% CO_2_ in the air. After 48 h, non-adherent cells were discarded, and adherent cells were replenished with a new culture medium. These freshly cultured cells were expanded with three-to-five changes of passage. T-MSC expanded over 5 passages were cryopreserved in a −200 °C liquid nitrogen (LN 2) tank using StrataCooler Cryo preservation modules (Agilent Technologies, Santa Clara, CA, USA). For freezing, T-MSCs grown in culture dishes were washed with phosphate-buffered saline (PBS, pH 7.4) and resuspended in a cryogenic medium containing 50% FBS, 40% DMEM, and 10% dimethyl sulfoxide (DMSO). All T-MSC used in this study were between passages 7 and 9.

### 4.2. Differentiation into T-MSC-PTHCs

T-MSCs were subcultured in DMEM-high glucose supplemented with 10% FBS. The cells were seeded either at low (50% of confluence) or high (100% of confluence) cell density and incubated in DMEM–high glucose supplemented with 10% FBS in the presence of activin A (100 ng/mL; R&D Systems Inc., Minneapolis, MN, USA) and sonic hedgehog (Shh) (100 ng/mL; R&D Systems) for 7 days. The medium was changed every 3 days. We used T-MSCs obtained from 3 donors to evaluate differentiation rates to parathyroid-like cells to take account of donor variation. After the differentiation, the cells were used for assessing gene and protein expressions, and the conditioned media (CM) were collected and tested for secreted PTH protein.

### 4.3. Droplet Digital PCR (dd PCR)

Total RNA was isolated using RNeasy (Qiagen, Hilden, Germany), and 2 ug of RNA was used for first-strand cDNA synthesis using the Superscript First-Strand Synthesis System (Invitrogen). A 20 uL mixture of PTH primer (F: 5′-aatggctgcgtaagaagctg-3′; R: 5′-agctttgtctgcctctccaa-3′), Bio-Rad 2X Evagreen supermix, and cDNA was emulsified with droplet generator oil (Bio-Rad, Hercules, CA, USA), using a QX-100 droplet generator according to the manufacturer’s instructions. The droplets were then transferred to 96-well reaction plates and heat sealed with pierceable sealing foil sheets. PCR amplification was performed in the sealed 96-well plates using a T100 thermocycler with the following cycling parameters: 10 min at 95 °C, 40 cycles with 30 s denaturation at 94 °C and a 60 s extension at 58 °C, followed by 10 min at 98 °C and hold at 12 °C. Immediately after PCR amplification, droplets were analyzed using a QX100 droplet reader (BioRad, Hercules, CA, USA), in which droplets from each well are aspirated, streamed toward a detector, and aligned for a single file two-color detection. Fluorescence data for each well were analyzed using Quantasoft software (Bio-Rad). Thresholds were manually set based on no-template control and negative control samples. Droplet positivity was determined by fluorescence intensity of samples, and PTH copies per uL were calculated by averaging over all the replicate wells. 

### 4.4. PTH ELISA

CM was collected 0 and 7 days of differentiation and stored at −80 °C until use. Each CM collected from the non-differentiated T-MSCs or differentiated T-MSC-PTHCs was prepared in triplicate and analyzed for concentrations of secreted PTH protein, using a human bioactive PTH 1-84 (Quidel #60-3000, Immutopics, San Clemente, CA, USA) enzyme-linked immunosorbent assay (ELISA) kit.

### 4.5. Western Blotting

The cultured cells were collected and lysed in RIPA lysis buffer (Sigma-Aldrich, St. Louis, MO, USA). Equal amounts of protein were separated by SDS-PAGE electrophoresis and transferred to the PVDF membrane (#ISEQ00010, Merck Millipore, Burlington, MA, USA). The following antibodies were used for Western blot analysis: mouse antibodies-PTH (1:500, #sc-69930, Santa Cruz Biotechnology, Santa Cruz, CA, USA), CaSR (1:2000, #MA1-934, Invitrogen Waltham, MA, USA), and GAPDH (1:3000, #LF-PA0212, AbFrontier, Seoul, Korea). 

### 4.6. Immunocytochemistry

For immunofluorescence microscopy, the cells were cultured on coverslips submerged in a 60 mm^2^ Petri dish. After the differentiation, the coverslip was collected, and the cells were fixed in 4% paraformaldehyde (#P6148, Sigma-Aldrich, St. Louis, MO, USA) for 30 min, permeabilized with 0.5% Triton X-100 (#TR1020-500, Biosesang, Sungnam, Korea) for 15 min, and then blocked with 5% BSA (# BSAS 0.1, Bovogen, Keilor East, Australia) prepared in PBS for 1 h then incubated with PTH antibody (1:200, LF-MA30443, AbFrontier, Seoul, Korea) for overnight at 4 °C. The coverslips were then incubated with secondary antibody (1:500, Alexa-488-conjugated goat anti-mouse IgG, # A-11001, Invitrogen, Waltham, MA, USA) for 1 h at room temperature. Visualization of images was carried out using a confocal microscope (LSM5 Pascall, Carl ZEISS, Oberkochen, Germany).

### 4.7. RNA Sequencing

Ion Torrent sequencing libraries were prepared according to the AmpliSeq Library prep kit protocol (Thermo Fisher, Waltham, MA, USA). A total of 50 ng of total RNA derived from T-MSCs or T-MSC-PTHCs of donor #1) was reverse transcribed, and the resulting cDNA was amplified for 11 cycles by adding PCR Master Mix and the AmpliSeq human transcriptome gene expression primer pool (over 20,000 human RefSeq genes). Amplicons were digested with the proprietary FuPa enzyme, and then, barcoded adapters were ligated onto the target amplicons. The library amplicons were bound to magnetic beads, and residual reaction components were washed off. The libraries were eluted and individually quantitated by qPCR using Ion Torrent P1, a sequencing primer, and TaqMan Probe master mix. Individual libraries were diluted to an 85 pM concentration, then combined in batches for further processing. Emulsion PCR, templating and 550 chip loading were performed with an Ion Chef Instrument (Thermo Fisher). Sequencing was performed on an Ion S5xl sequencer (Thermo Fisher) [33]. For human transcriptome analysis, the fold change in a normalized read count was determined for each replicate experiment, and the mean fold change was calculated for each gene. Genes with a mean fold change of >1.5 (T-MSCs vs. T-MSC-PTHCs) were analyzed using gene ontology enrichment analysis.

### 4.8. Animal Experiments

#### 4.8.1. PTX Rats

Male Sprague Dawley (SD) rats weighing 200 g (Orient Bio, Seongnam, Korea) were housed under a 12 h light-dark cycle and provided ad libitum access to a standard AIN-93G diet (calcium concentration ¼ 5 g/kg diet, 0.5%) before PTX. All the experimental procedures were reviewed and approved by the ethics committee for animal research at Ewha Woman’s University (ESM17-0393). Rat parathyroid glands were identified, and PTX was performed as previously described [34]. Briefly, male rats were photosensitized using 5-aminolevulinic acid hydrochloride (5-ALA, 500 mg/kg) via intraperitoneal injection and were kept under subdued light for 2 h to prevent phototoxic effects. After 2 h, rats were anesthetized with Zoletile (Virbac, Seoul, Korea)/Rompun (Bayer Korea, Seoul, Korea) (1:1 mixture, 0.1 mL/100 g body weight). The skin on the anterior part of the neck was longitudinally incised, exposing the trachea and thyroid. Parathyroid glands were detected under a xenon light source (405 nm) based on the presence of red fluorescence anterolaterally to the thyroid gland and bilateral parathyroid glands were surgically removed. After surgery, the incision was closed with 4-0 Ethilon^®^ sutures (Johnson & Johnson, New Brunswick, NJ, USA) and treated with betadine solution (5% Sterile Ophthalmic Prep Solution, Woodstock, IL, USA) to prevent infection of the wound area.

#### 4.8.2. Transplantation

Male SD rats were randomly allocated into 4 groups—2 without cell implantation (non-PTX, HA only) and 2 with cell (T-MSC-HA or T-MSC-PTHC-HA) implantation. The non-PTX control group was untreated and age-matched rats. Cells were mixed in 1.0 mL of a cross-linked hyaluronic acid filler (HA; Tissuefill, JW Pharmaceutical Co, Seoul, Korea) and 2.0 mL of culture media. Subsequently, cells and HA mixture were placed into a 50 mL centrifuge tube under sterile conditions. Once mixed, cell-HA (T-MSC-HA or T-MSC-PTHC-HA) mixture was loaded into a 1 mL syringe immediately prior to injection. All rats used in the study were given AIN-93G diet. The T-MSC-HA or T-MSC-PTHC-HA group received a subcutaneous injection of 1,000,000 cells in a volume of 200 uL per animal. PTH was measured by collecting blood at regular intervals for 16 weeks from mice in all experimental groups.

#### 4.8.3. Assessment of iPTH

Whole blood of PTX rats was collected by jugular vein puncture during 0–16 weeks after transplantation, and then, serum was separated by centrifugation and stored at −80 °C until use for iPTH concentrations. The iPTH in serum was measured using an intact PTH ELISA kit. 

### 4.9. Statistical Analysis

All statistical analyses were performed using SPSS ver. 21 (SPSS Inc., Chicago, IL, USA), and data were expressed as the mean ± standard deviation (S.D.), with n indicating the number of experiments. Statistical significance of differences was determined using a Student’s *t*-test for paired data, and one-way analysis of variance (ANOVA) for more than two pairs. The level of significance was represented as *p* < 0.05 (*), *p* < 0.01 (**), *p* < 0.001 (***) and *p* < 0.0001 (****). Different letters indicate significant differences among experimental groups (*p* < 0.05).

## Figures and Tables

**Figure 1 ijms-23-00715-f001:**
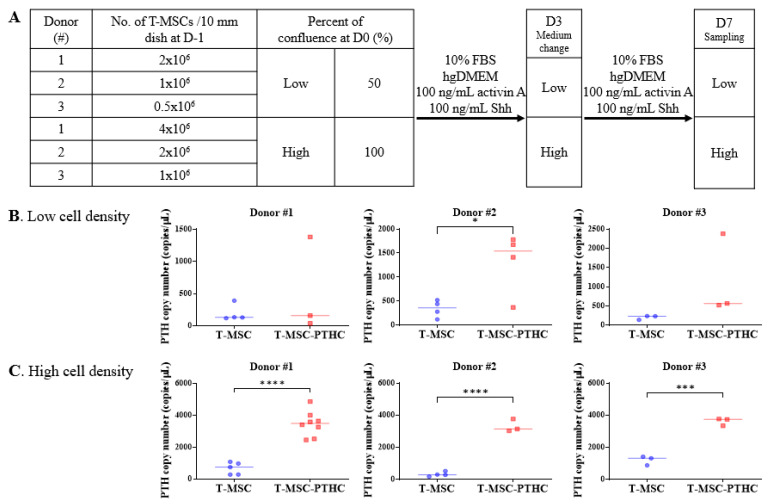
The effect of cell density on differentiation into PTH-releasing cells (PTHCs) from T-MSCs: (**A**) schematic representation of the differentiation protocol into the T-MSC-PTHCs. T-MSC seeded at a low cell density (50% confluence, and (**B**) a high cell density (100% confluence), (**C**) for differentiation. The number of PTH mRNA copies of T-MSC and T-MSC-PTHC from three donors were quantified and compared, respectively, using digital droplet PCR. Data are presented as mean ± SD from at least three independent experiments. The statistical analyses were performed using Student’s *t*-test (* *p* < 0.05, *** *p* < 0.001, **** *p* < 0.0001). PTH gene expression was significantly more increased in high cell density (**C**) than in low cell density (**B**) from all donors (#1~3); (**D**) the images of all the wells from 3 donors showing 50% and 100% confluence on day 0 (D0) and Day 7 (D7) (×40). Abbreviations: PTH, parathyroid hormone; T-MSC, tonsil-derived mesenchymal stem cells; T-MSC-PTHCs, T-MSC-derived PTH-releasing cells; Shh, Sonic hedgehog.

**Figure 2 ijms-23-00715-f002:**
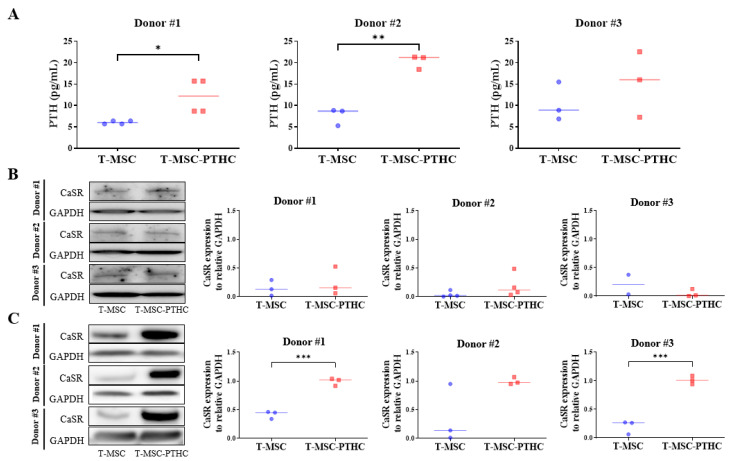
Establishment of the PTH-secretion potential of T-MSC-PTHCs from three donors: (**A**) conditioned medium was taken each step of T-MSC and T-MSC-PTHCs, and the concentration of PTH was measured using a commercial PTH ELISA kit at high cell density. Data are presented as the mean ± SD of at least three experiments (* *p* < 0.05, ** *p* < 0.01). PTH concentrations were significantly increased in T-MSC-PTHCs, compared with T-MSCs derived from donors #1 and #2, respectively. The expression of CaSR protein during differentiation of T-MSCs into T-MSC-PTHCs was measured by Western blotting and quantified using ImageJ software at low (**B**) and high (**C**) cell densities. Protein levels are normalized to GAPDH. Data are presented as the mean ± SD of at least three experiments (*** *p* < 0.001). CaSR expression levels were significantly increased in T-MSC-PTHCs, compared with T-MSCs derived from donors #1 and #3, respectively. Abbreviations: PTH, parathyroid hormone; T-MSC, tonsil-derived mesenchymal stem cells; T-MSC-PTHCs, T-MSC-derived PTH-releasing cells; CaSR, calcium-sensing receptor: GAPDH, glyceraldehyde 3-phosphate dehydrogenase.

**Figure 3 ijms-23-00715-f003:**
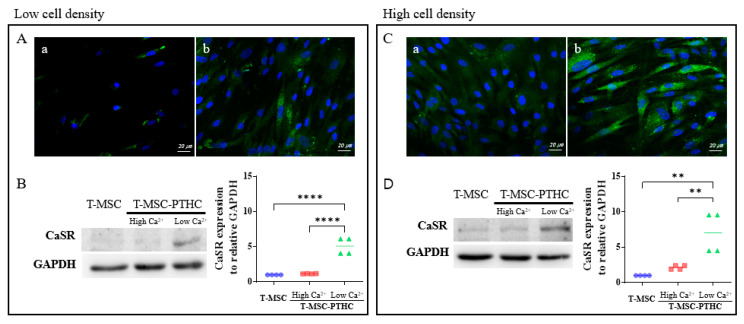
Comparison of differentiation potential of T-MSC-PTHCs according to cell density during differentiation (low cell density: A, B; high cell density: (**C**,**D**)): (**A**,**C**) the expression of PTH (green) in T-MSC (panel a) and T-MSC-PTHCs (panel b) derived from donor #1 was evaluated by immunostaining. The cells were counterstained with DAPI (blue). The expression of PTH of T-MSC-PTHCs was higher in high-density cells (**C**) than low-density cells (**A**); (**B**,**D**) expression of CaSR protein according to extracellular calcium level in T-MSC-PTHCs from three donor #1. T-MSC-PTHCs were further exposed to culture media containing low (0.09 mM) or high (3.0 mM) calcium concentrations for up to 48 h. The expression of CaSR was measured by Western blotting and quantified using ImageJ software and are normalized to GAPDH. Data are presented as the mean ± SD of at least three experiments (** *p* < 0.01, **** *p* < 0.0001). In both densities of cells, the expression of CaSR protein was significantly increased under exposure to low calcium and was not significantly increased under exposure to high calcium. Abbreviations: PTH, parathyroid hormone; T-MSC, tonsil-derived mesenchymal stem cells; T-MSC-PTHCs, T-MSC-derived PTH-releasing cells; CaSR, calcium-sensing receptor; GAPDH, glyceraldehyde 3-phosphate dehydrogenase.

**Figure 4 ijms-23-00715-f004:**
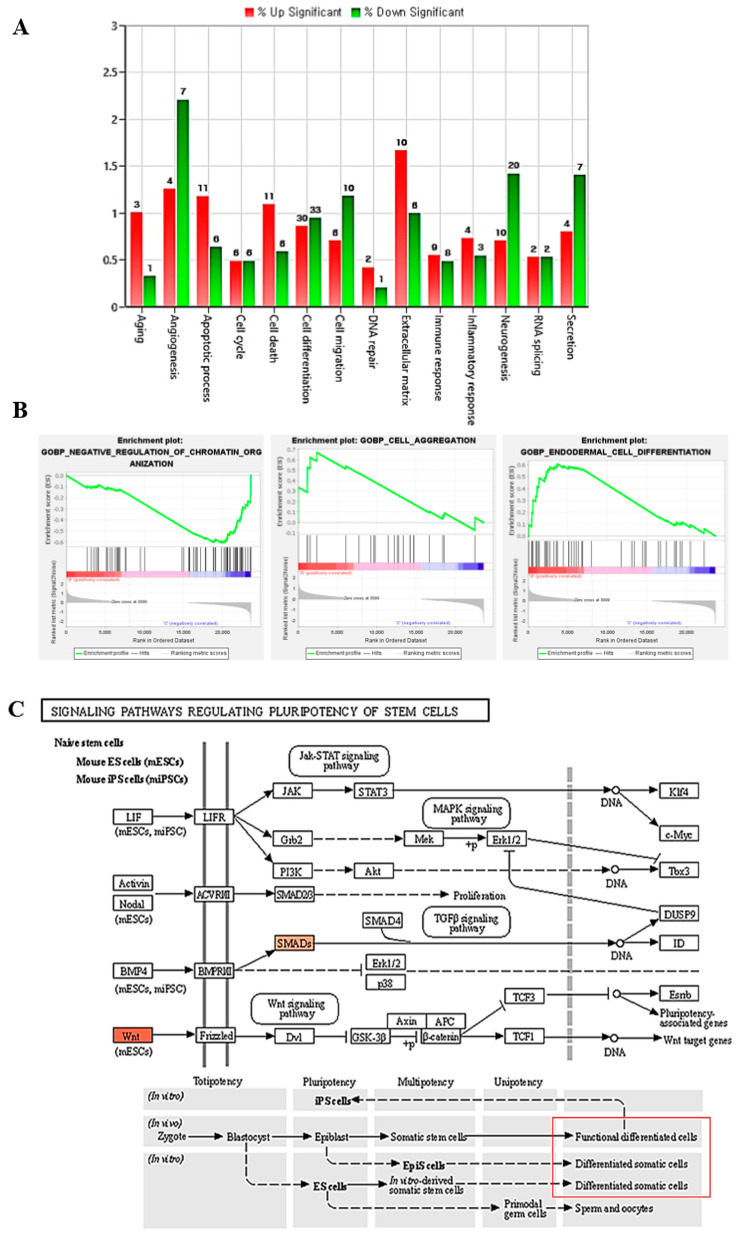
RNAseq analysis by gene set enrichment analysis (GSEA) and KEGG in T-MSCs versus T-MSC-PTHCs: (**A**) distribution of the genes of interest that were profiled; (**B**) GSEA plots for pathways involved in the three functional pathways; KEGG pathway analysis of regulating pluripotency of stem cell (**C**) and TGF-beta signal pathways (**D**). In orange are the upregulated transcripts.

**Figure 5 ijms-23-00715-f005:**
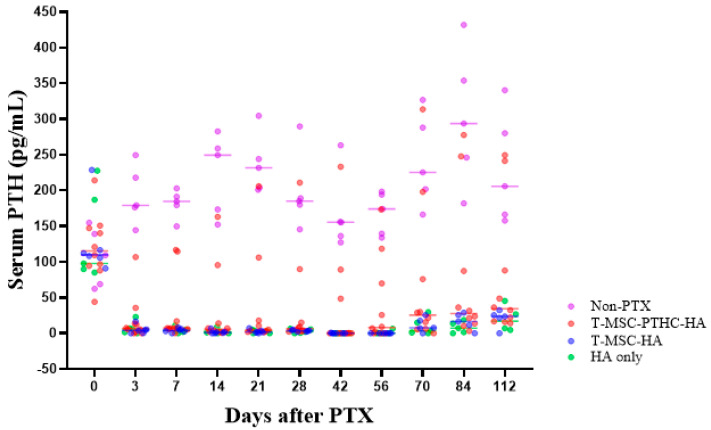
Restoration of intact PTH secretion in the rat models by transplantation of T-MSC-PTHCs. To observe the therapeutic effect of T-MSC-PTHCs, a PTX rat model in which parathyroid tissue was completely surgically removed was used. Transplantation of those cells derived from high cell-density donor #1 was performed by mixing with HA to increase the rate of engraftment in vivo. The concentration of intact PTH (iPTH) in the blood of the rats increased steadily and significantly from 1 week after transplantation in only the TMSC-PTHC-HA group, and the other experimental groups slightly increased after 8 weeks, but it was not significant. iPTH level for each group was measured using ELISA. Data are presented as the mean ± SD of at least three experiments. This animal experiment was performed by classifying it into a positive control group (untreated and age-matched rats: non-PTX), negative control group (HA only), experimental groups (cell with HA transplantation group: T-MSC-HA; T-MSC-PTHC-HA). Abbreviations: PTH, parathyroid hormone; T-MSC, tonsil-derived mesenchymal stem cells; T-MSC-PTHC, T-MSC-derived PTH-releasing cells; PTX, parathyroidectomy; HA, hyaluronic acid.

**Table 1 ijms-23-00715-t001:** List of upregulated and downregulated cell cycle-related genes in T-MSC-PTHCs.

	Genes	Transcript_id	Description	Fold Change of T-MSC-PTHC (per T-MSC)
Upregulated genes	EZR	NM_003379	ezrin	1.961
	TPX2	NM_012112	TPX2, microtubule-associated	1.748
	TDRKH	NM_006862	tudor and KH domain containing	1.704
	MDM2	NM_001145339	MDM2 proto-oncogene, E3 ubiquitin protein ligase	1.607
	TAF10	NM_006284	TATA-box binding protein associated factor 10	1.588
	KIFC2	NM_145754	kinesin family member C2	1.522
Downregulated genes	RGCC	NM_014059	regulator of cell cycle	0.624
	FGF10	NM_004465	fibroblast growth factor 10	0.613
	PARD3	NM_001184789	par-3 family cell polarity regulator	0.604
	PPME1	NM_016147	protein phosphatase methylesterase 1	0.603
	CENPC	NM_001812	centromere protein C	0.585
	SDCCAG8	NM_006642	serologically defined colon cancer antigen 8	0.555

**Table 2 ijms-23-00715-t002:** List of upregulated and downregulated differentiation-related genes in T-MSC-PTHCs.

	Genes	Transcript_id	Description	Fold Change of T-MSC-PTHC (per T-MSC)
Upregulated genes	WNT2	NR_024047	Wnt family member 2	2.881
	EFNB2	NM_004093	ephrin B2	2.769
	LPAR1	NM_057159	lysophosphatidic acid receptor 1	2.591
	COL1A1	NM_000088	collagen type I alpha 1	2.202
	TAGLN	NM_003186	transgelin	2.186
	ALPL	NM_000478	alkaline phosphatase, liver /bone /kidney	2.045
	CTHRC1	NM_138455	collagen triple helix repeat containing 1	1.986
	PTX3	NM_002852	pentraxin 3	1.976
	BCL6	NM_001706	B-cell CLL /lymphoma 6	1.962
	EZR	NM_003379	ezrin	1.961
Downregulated genes	IL34	NM_001172771	interleukin 34	0.525
	HMGB2	NM_002129	high mobility group box 2	0.517
	STXBP1	NM_003165	syntaxin binding protein 1	0.509
	AKR1C1	NM_001353	aldo-keto reductase family 1, member C1	0.505
	GAP43	NM_002045	growth-associated protein 43	0.482
	KITLG	NM_003994	KIT ligand	0.466
	TMEM100	NM_001099640	transmembrane protein 100	0.418
	ANXA2	NM_004039	annexin A2	0.410
	TMEFF2	NM_016192	transmembrane protein with EGF like and two follistatin like domains 2	0.365
	NPTX1	NM_002522	neuronal pentraxin 1	0.310

## Data Availability

The RNAseq data used in this study are openly available in reference number [GSE193108] in the GEO database.

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
