# Peer review of "Density-Dependent Differentiation of Tonsil-Derived Mesenchymal Stem Cells into Parathyroid-Hormone-Releasing Cells"

_ijms, 2022, doi:10.3390/ijms23020715_

Round 1

Reviewer 1 Report

In this manuscript, the authors studied the way to optimize the differentiate rate in to parathyroid-hormone-releasing cells, which can restore the parathyroid functions. The differentiation efficiency varied depending on the cells from specific donor. In addition, the differentiation rate of the cells could be improved at a high cell density. The authors also detected RNA profile to identify the cell differentiation. Their findings contributed a differentiation protocol for the clinical application of tonsil-derived mesenchymal stem cells therapy for the treatment of hypothyroidism. 

It is even better to increase the number of donor.

Author Response

In this manuscript, the authors studied the way to optimize the differentiate rate in to parathyroid-hormone-releasing cells, which can restore the parathyroid functions. The differentiation efficiency varied depending on the cells from specific donor. In addition, the differentiation rate of the cells could be improved at a high cell density. The authors also detected RNA profile to identify the cell differentiation. Their findings contributed a differentiation protocol for the clinical application of tonsil-derived mesenchymal stem cells therapy for the treatment of hypoparathyroidism. 

It is even better to increase the number of donors.

Response: Thank you very much for your positive review. Your comments to our manuscript gave us insight for the development of tonsil-derived mesenchymal stem cells for clinical use.

For the number of donors, we selected 3 representative donor samples among 6 based on their proliferation and differentiation abilities. I am sorry we could not use T-MSCs cell line from 6 donors. Experiments with 6 donor samples were too massive to deal in our lab.

Reviewer 2 Report

In the manuscript, the authors reported the effect of cell density in the differentiation of tonsil-derived MSCs into PTH-releasing cells. Currently, there are significant and minor areas of concern for the current study that are outlined below.

Major concerns.

  1. The authors' main observation is based on density-dependent differences in cell differentiation. However, the experimental scheme seemed problematic as authors, instead of using fixed cell numbers, using different cell numbers for different donors. This experimental setup is not appropriate for concluding cell density-based observations. Authors need to perform experiments where all those cell densities are used for each donor (examples, 0.5 million, 1 million, 2 million, 4 million each for donor 1, 2, and 3). If all these cells were from humans, how come all those different numbers of cells gave perfectly 50% or 100% confluence.
  2. The authors did not provide any details on how the cells were collected, stored, cultured, harvested, which passages were used. Providing a reference (33) that does not exist (in the manuscript, only 31 references are provided) raises the serious question of how dependable all these data are. Authors need to provide detailed descriptions of the relevant details.   
  3. Authors need to provide the (brightfield) images for all the wells for 3 donors showing 50% and 100% confluence on day 0 and Day 7.
  4. Authors need to present all the data points (as dot plots) for all the graphs that are currently presented as bar graphs to give a better representation of each measurement.
  5. As the authors' main observation is cell density-dependent differentiation changes, the authors need to present data from both (low and high) density experiments side by side for PTH and CaSR.
  6. Cell differentiation is often known to work better when cells are in close contact and near confluent or completely confluent and previously reported for many other cell types and thus is not a major new observation.
  7. The authors did not present the data from their RNAseq experiment with adequate details with the experimental conditions and used samples used for the transcriptomics profiling.  
  8. The authors need to make the raw data for the RNAseq experiments available (by depositing the RAW and analyzed data in the GEO database) and provide the GSE number listed in the manuscript. In addition, authors also need to include the complete list of differentially expressed genes with the fold change, p-values, and other key measures as supplementary tables and top 10 or 20 up and down-regulated as part of the main manuscript.
  9. The quality/resolution of Figure 4B is not adequate, making it completely worthless. It is currently not readable, and none of the related conclusions can be confirmed.  
  10. Why were the parathyroid secretion-related pathway/s not enriched in the T-MSC-PTHC? To conclude, the T-MSC-PTHC authors need to do a better job analyzing and interpreting the transcriptomics data to support their conclusions further so that their RNA-seq data could confirm their conclusion about the functionality of these cells.

Minor concerns –

  • It is not clear what is the non-PTX control.
  • Authors need to provide details about their experimental conditions. For example, for 4.6, Immunocytometry authors need to give more details for their staining conditions (examples, antibody clone details, catalog number, dilution used). Should provide all the reagent details (Reagent name, Company, Catalog number, any other details) as lists (Tables) and need to be included as Supplementary data.

Author Response

In the manuscript, the authors reported the effect of cell density in the differentiation of tonsil-derived MSCs into PTH-releasing cells. Currently, there are significant and minor areas of concern for the current study that are outlined below.

Major concerns.

  1. The authors' main observation is based on density-dependent differences in cell differentiation. However, the experimental scheme seemed problematic as authors, instead of using fixed cell numbers, using different cell numbers for different donors. This experimental setup is not appropriate for concluding cell density-based observations. Authors need to perform experiments where all those cell densities are used for each donor (examples, 0.5 million, 1 million, 2 million, 4 million each for donor 1, 2, and 3). If all these cells were from humans, how come all those different numbers of cells gave perfectly 50% or 100% confluence.

Response: Yes, we agreed to the reviewer’s opinion. We added more sentences for the background and more illustrations for the experimental scheme.

Primary goal of our experiment was overcome donor-dependent variation of differentiation. The proliferation rate and differentiation ability of T-MSCs showed donor-to donor variation. The donor-to donor variation was a big barrier to us. That’s why we tried to start differentiation T-MSCS into PTH-releasing cells when proliferation T-MSCs reached confluency. Reaching 50% or 100% of confluency of each T-MSC lines were different due to donor-dependent variation.

To make clarify, we revised our manuscript as following,

In Introduction, page 2, line 69-71, new sentences were added.

However, proliferation rate and differentiation ability of T-MSCs showed donor-to donor variation. This donor-dependent variation of T-MSCs gives unexpected limitations to preclinical and clinical development of T-MSCs.  

In results, page 3, lines 99-103, new sentences were added and revised.

The proliferation capacity of T-MSCs from three donors showed donor-to-donor variation. Therefore, the cell numbers to reach 50 and/or 100% confluency were different from each donor. The scheme in Figure 1A illustrates the protocol for differentiation into T-MSC-PTHCs (D7). The approximate number of cells required to reach 50 or 100% confluency in one day for each donor are provided in the scheme shown in Figure 1A.

In results, page 3, lines 111-114, new sentences were added.

cells at D0 and D7 used in this study were observed by a light microscope (Figure 1D). The images of the T-MSCs from the donor #1 at high cell density showed the most densely packed cells in the cell culture dish.

Newly added Figure 1. (D) The images of all the wells from 3 donors showing 50% and 100% confluence on day 0 (D0) and Day 7 (D7) (x40).

In discussion, page 9, line 243-245, new sentences were added.

Donor-dependent variation in proliferation and differentiation of mesenchymal stem cells have been also observed in other mesenchymal stem cells from different tissue sources, such as bone marrow, adipose tissue and umbilical cord blood [20-22].

As consequences of this revision 3 new references have been added.

  1. Phinney, D. G.; Kopen, G.; Righter, W.; Webster. S.; Tremain, N.; Prockop, D. Donor variation in the growth properties and osteogenic potential of human marrow stromal cells. J. Cell Biochem. 1999, 75, 424–436. https://doi.org/10.1002/(SICI)1097-4644(19991201)75:3<424::AID-JCB8>3.0.CO;2-8
  2. Siddappa, R.; Licht, R.; van Blitterswijk, C.; de Boer, J. Donor variation and loss of multipotency during in vitro expansion of human mesenchymal stem cells for bone tissue engineering. Orthop. Res. 2007, 25, 1029–104. doi: 10.1002/jor.20402.
  3. Kang, I.; Lee, B.; Choi, S.W.; Lee, J.Y.; Kim, J.J.; Kim, B.E.; Kim, D.H.; Lee, S.E.; Shin, N.; Seo, Y.; Kim, H.S.; Kim, D.I.; Kang, K.S. Donor-dependent variation of human umbilical cord blood mesenchymal stem cells in response to hypoxic preconditioning and amelioration of limb ischemia. Exp Mol Med. 2018, 50, 1–15. https://doi.org/10.1038/s12276-017-0014-9.

  1. The authors did not provide any details on how the cells were collected, stored, cultured, harvested, which passages were used. Providing a reference (33) that does not exist (in the manuscript, only 31 references are provided) raises the serious question of how dependable all these data are. Authors need to provide detailed descriptions of the relevant details.

Response: We are sorry. We revised the manuscript according to the comments. We described in detail in the materials and methods (page 11). The descriptions were rewritten with matching the references (6, 7 and 18).

T-MSC isolation and culture were conducted as previously described [6,7,18].

To establish T-MSCs from tonsillar tissues, isolated tonsillar tissue was minced and digested in Dulbecco’s modified Eagle medium (DMEM; Invitrogen, Carlsbad, California) containing 210 U/ml collagenase type I (Invitrogen, Carlsbad, CA, USA) and 10 lg/ml DNase (Sigma-Aldrich, St. Louis, Missouri) for 30 min at 37 °C. The digested tissue was filtered through a wire mesh and washed with RPMI-1640 medium, after which adherent mononuclear cells were obtained by Ficoll-Paque (GE Healthcare, Little Chalfont, UK) density gradient centrifugation. The cells were cultured for 48 hours at 37 °C in high-glucose (4500 mg/L) DMEM containing 10% fetal bovine serum (FBS; Invitrogen) and 1% penicillin/ streptomycin (Sigma-Aldrich) in a humidified chamber under 5% CO2 in the air. After 48 h, non-adherent cells were discarded and adherent cells were replenished with a new culture medium. These freshly cultured cells were expanded with three to five changes of passage. T-MSC expanded over 5 passages were cryopreserved in a -200 °C liquid nitrogen (LN 2) tank using StrataCooler Cryo preservation modules (Agilent Technologies, Santa Clara, CA, USA). For freezing, T-MSCs grown in culture dishes were washed with phosphate-buffered saline (PBS, pH 7.4) and resuspended in a cryogenic medium containing 50% FBS, 40% DMEM and 10% dimethyl sulfoxide (DMSO). All T-MSC used in this study were between passages 7 and 9.

  1. Authors need to provide the (brightfield) images for all the wells for 3 donors showing 50% and 100% confluence on day 0 and Day 7.

Response: Yes, we provided the brightfield images for all the wells for 3 donors in 50% and 100% confluence on day 0 (D0) and day 7 (D7) in the figure 1D.

Figure 1. (D) The images of all the wells from 3 donors showing 50% and 100% confluence on day 0 (D0) and Day 7 (D7) (x40).

  1. Authors need to present all the data points (as dot plots) for all the graphs that are currently presented as bar graphs to give a better representation of each measurement.

Response: Yes, we converted to the dot plots with an overlay of the raw data from the bar graphs in every figure.

  1. As the authors' main observation is cell density-dependent differentiation changes, the authors need to present data from both (low and high) density experiments side by side for PTH and CaSR.

Response: Yes, we provided the data from low density experiments for CaSR.

According to reviewer’s comments, we revised the manuscript as following,

In results, page 4, lines 125-131,

The high cell density group from all donors had a higher differentiation efficiency than the low cell density in the ddPCR analysis, and the secreted PTH concentration and protein level of calcium-sensing receptor (CaSR) were measured using ELISA and western blots, respectively. In the case of the T-MSCs at low cell density, PTH secretion was not significantly different between pre- and post- differentiation (data not shown). The PTH concentration increased only in all donors of high cell density after differentiation, but in particular, donors #1 and #2 showed a significant increase (p<0.05 and p<0.01, respectively) (Figure 2A).

Page 4, line 139-140,

The expression of CaSR protein did not significantly increase at low cell density (Figure 2B) and

We added new images in Figure 2(B), page 5,

Figure 2. Establishment of the PTH-secretion potential of T-MSC-PTHCs from three donors. (A) Conditioned medium was taken each step of T-MSC and T-MSC-PTHCs, and the concentration of PTH was measured using a commercial PTH ELISA kit at high cell density. Data are presented as the mean ± SD of at least three experiments (*p<0.05, **p<0.01). PTH concentration were significantly increased in T-MSC-PTHCs than in T-MSCs derived from donors #1 and #2, respectively. The expression of CaSR protein during differentiation of T-MSCs into T-MSC-PTHCs was measured by Western blot and quantified using ImageJ software at low (B) and high (C) cell density. Protein levels are normalized to GAPDH. Data are presented as the mean ± SD of at least three experiments (***p<0.001). CaSR expression were significantly increased in T-MSC-PTHCs than in T-MSCs derived from donors #1 and #3 respectively. Abbreviations: PTH, parathyroid hormone; T-MSC, tonsil-derived mesenchymal stem cells; T-MSC-PTHCs, T-MSC-derived PTH-releasing cells; CaSR, calcium-sensing receptor: GAPDH, glyceraldehyde 3-phosphate dehydrogenase.

  1. Cell differentiation is often known to work better when cells are in close contact and near confluent or completely confluent and previously reported for many other cell types and thus is not a major new observation.

Response: Yes, we respect your opinion. Therefore, the phrase that the differentiation method of this study is a major new observation has been deleted.

  1. The authors did not present the data from their RNAseq experiment with adequate details with the experimental conditions and used samples used for the transcriptomics profiling.

Response: Yes, I have revised the manuscript as you pointed out.

We analyzed again for revision.

In results, pages 6, we added and revised as following,

Top 20 most significantly upregulated and downregulated expressed genes of the T-MSC-PTHCs were shown in supplemental tables 1 and 2, respectively. 12 and 20 known genes related to cell cycle and differentiation were selected respectively, which were significantly changed in T-MSC-PTHCs compared with the undifferentiated T-MSCs (Tables 1 and 2). Clusters of cell cycle and differentiation were found only within a minority of the T-MSC-PTHCs (Figure 4A). Changes in the expression of genes were involved in the active pathways of the T-MSC-PTHCs. For example, the genes involved in the negative regulation of chromatin organization, cell aggregation and endodermal differentiation were significantly upregulated in the T-MSC-PTHC group (Figure 4B). Based on the transcriptome Analysis Console (TAC) results, terms related to “signaling pathway regulating pluripotency of stem cells” for functional differentiated cells and “TGF-beta signaling” for G1 arrest of cell cycle in the KEGG pathways, Wnt, SMADs, BMP and Smad1/5/8 were increased in T-MSC-PTHCs (Figure 4C).

In materials and methods, page 12,

Ion Torrent sequencing libraries were prepared according to the AmpliSeq Library prep kit protocol (Thermo Fisher). A total of 50 ng of total RNA derived from T-MSCs or T-MSC-PTHCs of donor #1) was reverse transcribed, and the resulting cDNA was amplified for 11 cycles by adding PCR Master Mix and the AmpliSeq human transcriptome gene expression primer pool (over 20,000 human RefSeq genes). Amplicons were digested with the proprietary FuPa enzyme, and then, barcoded adapters were ligated onto the target amplicons. The library amplicons were bound to magnetic beads, and residual reaction components were washed off. The libraries were eluted and individually quantitated by qPCR using Ion Torrent P1, a sequencing primer, and TaqMan Probe master mix. Individual libraries were diluted to an 85 pM concentration, then combined in batches for further processing. Emulsion PCR, templating and 550 chip loading were performed with an Ion Chef Instrument (Thermo-Fisher). Sequencing was performed on an Ion S5xl sequencer (Thermo-Fisher) [33]. For human transcriptome analysis, the fold change in a normalized read count was determined for each replicate experiment, and the mean fold change was calculated for each gene. Genes with a mean fold change of >1.5 (T-MSCs vs. T-MSC-PTHCs) were analyzed using gene ontology enrichment analysis.

  1. The authors need to make the raw data for the RNAseq experiments available (by depositing the RAW and analyzed data in the GEO database) and provide the GSE number listed in the manuscript.

Response: We analyzed again for revision. After finishing analyze the data, the raw data has been submitted to GEO database. We are now waiting for finalizing the submission.

We hope we can add GSE number soon before finalize review of this manuscript.  

Data available in a publicly accessible repository: The RNAseq data used in this study are openly available in reference number [xxxxxxx] in the GEO database.

 The reply email from GEO database has been attached below,

Genes

transcript_id

Description

Fold change of T-MSC-PTHC (per T-MSC)

Up-

regulated genes

EZR

NM_003379

ezrin

1.961

TPX2

NM_012112

TPX2, microtubule-associated

1.748

TDRKH

NM_006862

tudor and KH domain containing

1.704

MDM2

NM_001145339

MDM2 proto-oncogene, E3 ubiquitin protein ligase

1.607

TAF10

NM_006284

TATA-box binding protein associated factor 10

1.588

KIFC2

NM_145754

kinesin family member C2

1.522

Down-

regulated genes

RGCC

NM_014059

regulator of cell cycle

0.624

FGF10

NM_004465

fibroblast growth factor 10

0.613

PARD3

NM_001184789

par-3 family cell polarity regulator

0.604

PPME1

NM_016147

protein phosphatase methylesterase 1

0.603

CENPC

NM_001812

centromere protein C

0.585

SDCCAG8

NM_006642

serologically defined colon cancer antigen 8

0.555

In addition, authors also need to include the complete list of differentially expressed genes with the fold change, p-values, and other key measures as supplementary tables and top 10 or 20 up and down-regulated as part of the main manuscript.

Response: Yes, we added tables to the manuscript and supplementar data, as you pointed out.

In results, pages 6-7,

Top 20 most significantly upregulated and downregulated expressed genes of the T-MSC-PTHCs were shown in supplemental tables 1 and 2, respectively.

Table 1. List of upregulated and downregulated cell cycle-related genes in the T-MSC-PTHCs.

Table 2. List of upregulated and downregulated differentiation-related genes in the T-MSC-PTHCs.

Genes

transcript_id

Description

Fold change of T-MSC-PTHC (per T-MSC)

Up-

regulated genes

WNT2

NR_024047

Wnt family member 2

2.881

EFNB2

NM_004093

ephrin B2

2.769

LPAR1

NM_057159

lysophosphatidic acid receptor 1

2.591

COL1A1

NM_000088

collagen type I alpha 1

2.202

TAGLN

NM_003186

transgelin

2.186

ALPL

NM_000478

alkaline phosphatase, liver /bone /kidney

2.045

CTHRC1

NM_138455

collagen triple helix repeat containing 1

1.986

PTX3

NM_002852

pentraxin 3

1.976

BCL6

NM_001706

B-cell CLL /lymphoma 6

1.962

EZR

NM_003379

ezrin

1.961

Down-

regulated genes

IL34

NM_001172771

interleukin 34

0.525

HMGB2

NM_002129

high mobility group box 2

0.517

STXBP1

NM_003165

syntaxin binding protein 1

0.509

AKR1C1

NM_001353

aldo-keto reductase family 1, member C1

0.505

GAP43

NM_002045

growth associated protein 43

0.482

KITLG

NM_003994

KIT ligand

0.466

TMEM100

NM_001099640

transmembrane protein 100

0.418

ANXA2

NM_004039

annexin A2

0.410

TMEFF2

NM_016192

transmembrane protein with EGF like and two follistatin like domains 2

0.365

NPTX1

NM_002522

neuronal pentraxin 1

0.310

  1. The quality/resolution of Figure 4B is not adequate, making it completely worthless. It is currently not readable, and none of the related conclusions can be confirmed.

Response: Yes, we analyzed again and made new images for the transciptome analysis. 

In results, figure 4 has been complete revised, 

Figure 4. RNA-Seq analysis by Gene Set Enrichment Analysis (GSEA) and KEGG in T-MSCs versus T-MSC-PTHCs. (A) Distribution of the genes of interested that were profiled. (B) GSEA plot for pathways involved in the three functional pathways. (C) KEGG pathway analysis of regulating pluripotency of stem cell and TGF-beta signal pathways. In orange are the up-regulated transcripts.

  1. Why were the parathyroid secretion-related pathway/s not enriched in the T-MSC-PTHC? To conclude, the T-MSC-PTHC authors need to do a better job analyzing and interpreting the transcriptomics data to support their conclusions further so that their RNA-seq data could confirm their conclusion about the functionality of these cells.

Response: Yes, we respect your opinion. Therefore, we performed re-analysis and presented additional results indicating that T-MSC-PTHCs is related to the pathway of differentiation into endodermal cell.

 Some part of revision according to this comment has been written in responses for comment 7.

In discussion, page 10, lines 282-289,

RNAseq analysis showed that cell cycle and differentiation signal pathway were found only within a minority in T-MSC-PTHCs. In particular, GSEA results showed that negative regulation of chromatin organization, cell aggregation and endodermal differentiation signaling pathway enriched with genes were upregulated in T-MSC-PTHCs. KEGG pathways analysis indicated that Wnt, SMADs, BMP and Smad1/5/8 relating to pluripotency of stem cell pluripotency-regulating pathway and G1 arrest of cell cycle in the TGF-beta signaling pathway as well as endodermal differentiation were increased in the T-MSC-PTHCs.

Minor concerns –

  • It is not clear what is the non-PTX control.

Response: Yes, I have corrected it in the manuscript as you pointed out.

In Methods and Materials, page 13,

The non-PTX control group was non-treated and age-matched rats.

  • Authors need to provide details about their experimental conditions. For example, for 4.6, Immunocytometry authors need to give more details for their staining conditions (examples, antibody clone details, catalog number, dilution used). Should provide all the reagent details (Reagent name, Company, Catalog number, any other details) as lists (Tables) and need to be included as Supplementary data.

Response: Yes, I have corrected it in the manuscript as you pointed out. Methods and materials were revised including catalog numbers though the section.

Thank you very much for your valuable comments.
